# The Impact of Transitions in Caregiving Status on Depressive Symptoms among Older Family Caregivers: Findings from the Korean Longitudinal Study of Aging

**DOI:** 10.3390/ijerph18010042

**Published:** 2020-12-23

**Authors:** Kyungduk Hurh, Hin Moi Youn, Yoon Sik Park, Eun-Cheol Park, Sung-In Jang

**Affiliations:** 1Department of Preventive Medicine, Yonsei University College of Medicine, Seoul 03722, Korea; hkd4397@yuhs.ac (K.H.); SKYPARKYS2@yuhs.ac (Y.S.P.); ECPARK@yuhs.ac (E.-C.P.); 2Institute of Health Services Research, Yonsei University, Seoul 03722, Korea; moiyoun@yuhs.ac

**Keywords:** mental health, depressive symptoms, family caregivers, caregiving status, activities of daily living

## Abstract

This study identifies the effects of transitions in caregiving status on depressive symptoms among middle-aged or older adults who care for family members with limitations in activities of daily living (ADL). Data were collected from the 2006–2018 Korean Longitudinal Study of Aging. A total of 7817 subjects were included. On the basis of their caregiving status transition, participants were categorized into four groups: started caregiving, continued caregiving, stopped caregiving, and noncaregivers. Depressive symptoms were measured using the 10 item Center for Epidemiologic Studies Depression Scale. Analysis using a generalized estimating equation model and subgroup analyses were conducted. Compared to noncaregivers, women who started caregiving showed more depressive symptoms in the following year (β 0.761, *p* < 0.0001). Regardless of sex, older adults who continued caregiving had more depressive symptoms than noncaregivers did (β 0.616, *p* < 0.0277 in men, and β 1.091, *p* < 0.0001 in women). After relinquishing caregiving responsibilities to other caregivers, participants’ depressive symptoms in the following year showed no statistically significant difference from that of noncaregivers. Thus, starting or continuing caregiving was associated with increased depressive symptoms, and those symptoms could be normalized by stopping caregiving. Intervention strategies to reduce family caregivers’ depressive symptoms are needed.

## 1. Introduction

Depression is a serious and increasingly common global mental problem, with more than 160 million people affected in 2017 [1]. In South Korea, the number of individuals with major depressive disorder was 908,000 (1.78%) in 2019, an increase from 788,000 (1.69%) in 2009 [2]. Moreover, many more South Koreans, ranging from 25.3% to 38.9% of the total population, were reported to have depressive symptoms [3].

Many factors can affect the development of depressive symptoms, including predisposing genetic influences, exposure to traumatic events, adverse social factors, a past history of depression, and recent stressful life events and difficulties [4].

Difficulties with self-care in daily life can be stressful not only for the concerned individual, but also their family members. Family members who act as caregivers for patients with limitations in activities of daily living (ADL) are especially at higher risk of having depressive symptoms [5,6,7]. Since the dependence in ADL is associated with adverse health conditions, such as dementia or stroke, watching a loved one suffer over time may cause great psychological distress to family caregivers [8,9]. Moreover, the burden caused by other factors associated with caregiving, such as financial stress, social isolation, and lack of personal time, may further distress family caregivers [10].

Caregiver stress is the consequence of a complex process comprising a number of inter-related factors [11]. Previous studies suggest that the psychological consequence of caregiving can worsen by being female, having a lower income, being the patient’s spouse, spending long hours caregiving, and by the severe ADL dependency of the care receiver [12,13]. Transitions in caregiving status are also thought to have a differential effect on a caregiver’s depressive symptoms [14,15]. Typically, starting or continuously providing family caregiving is regarded to be associated with an increase in the caregiver’s depressive symptoms [16,17,18,19]. In terms of ceasing caregiving, there are mixed findings on the incidence of depressive symptoms. Several studies reported that the ceasing of caregiving, either through bereavement or institutionalization, has a positive effect on the family caregiver’s mental health [20,21]. By contrast, other studies suggest that family caregivers show increased depressive symptoms after bereavement [14,15,18,22,23].

In South Korea, particularly, the number of older adults with ADL limitations is expected to increase due to a rapidly aging population [24]. Moreover, in South Korea, the majority of older adults are cared for by their family members, who are also aged themselves [25]. Therefore, a better understanding of older caregivers’ depressive symptoms is needed to support these individuals. Numerous studies reported depressive symptoms of South Korean family caregivers who care for patients with stroke, cancer, dementia, and many other diseases [26,27,28]. However, to our best knowledge, no studies have evaluated the differential effects of transitions in and out of family caregiving on depressive symptoms among the South Korean population. This study, therefore, investigates the magnitude of depressive symptoms among caregivers on the basis of caregiving-status transition using large national longitudinal survey data. In addition, subgroup analyses were conducted to answer the following questions: After relinquishing caregiving responsibilities to others, how would a caregiver’s depressive symptoms change? What is the effect of gender and other sociodemographic factors such as age, residential area, current economic activity, and number of household members on the association between caregiving-status transitions and depressive symptoms?

## 2. Materials and Methods

### 2.1. Study Sample

Study data were collected from the Korea Longitudinal Study of Aging (KLoSA) for 2006, 2008, 2010, 2012, 2014, 2016, and 2018. The KLoSA is a nationally representative survey conducted every two years by the Korea Employment Information Service (KEIS). It aims to generate basic data needed to devise and implement policies that address emerging trends in the process of population aging. Details on the KLoSA can be found on the KEIS webpage [29]. In 2006, the original panel sample was composed of 10,254 adults aged 45 years and over (born in 1961 or earlier) who resided in South Korea. The retention rates of the survey sample were 86.6%, 81.7%, 80.1%, 80.4%, 79.6%, and 78.8% for the 2008, 2010, 2012, 2014, 2016, and 2018 surveys, respectively [30]. Participants who were already taking antidepressants at the time of the survey or those who had any incomplete data were excluded. The total number of participants was 7817 in the final sample of the baseline study year from 2006 to 2008. 

All data were stratified by sex. Of the total 7817 participants, 3314 (42.4%) were men and 4503 (57.6%) were women. We included age (45–64, 65–74, 75 or more), area of residence (metropolitan, urban, rural), education level (elementary school or lower, middle or high school, college or higher), economic activity (active, inactive), equivalized household income (divided into quartiles), marital status (with spouse, without spouse), number of household members (1, 2, 3 or more), and participation in social activities (yes, no) as demographic and socioeconomic variables. The mean and standard deviation of the participants’ ages were 65.1 ± 10.4 for men and 65.8 ± 11.0 for women. Lifetime smoking experience (yes, no), current alcohol drinking (yes, no), number of chronic diseases (none, 1, 2, or more), and participation in regular physical activity more than once a week (yes, no) were included as health-related factors. Chronic diseases include hypertension, diabetes, cancer, chronic obstructive pulmonary disease, liver disease, cardiac disease, psychiatric disorders, cerebrovascular disease, osteoarthritis, and rheumatoid diseases.

### 2.2. Instruments

To evaluate participants’ depressive symptoms, we used the 10 item Center for Epidemiologic Studies Depression Scale (CES-D 10), a truncated version of the original 20 item CES-D [31]. The CES-D 10 scores 0 or 1 for each of the 10 items, and total score ranges from 0 to 10. Example items are “During last week, I felt depressed” and “During last week, I felt that everything I did was an effort”; respondents could answer dichotomously. Higher overall scores suggested more severe depressive symptoms. The scale’s validity and reliability as a screening instrument for depressive symptoms in older adults were verified in previous studies [32,33]. The internal consistency of the CES-D 10 was satisfactory (Cronbach α = 0.81) in this study.

ADL can be defined as the basic ability to take care of oneself to independently perform complex daily activities and maintain a social life. ADL can be evaluated in two main ways: basic or instrumental ADL (IADL) [34]. “ADL” here refers to basic ADL, including dressing, washing one’s face, hair and tooth brushing, bathing and showering, eating, getting across a room, getting out of bed, using the toilet, and controlling urination or defecation [30]. In order to distinguish it from usual childcare, “family members requiring ADL assistance” included only those who were aged 10 or older [30].

A caregiving-status transition between two adjacent surveys was classified into one of four categories: (1) None, (2) Started, (3) Stopped, and (4) Continued. We conducted additional analysis and classified participants into nine categories on the basis of the presence of family members with ADL limitations and the participants’ caregiving status: The noncaregiver group was divided into four categories: (1) Absent → Absent (no family members with ADL limitations), (2) Absent → By others (participants having family members with ADL limitations who were cared for by other caregivers), (3) By others → Absent, and (4) By others → By others. The “started caregiving” group was divided into two categories: (5) Absent → By myself (family caregivers) and (6) By others → By myself. The “stopped caregiving” group was divided into two categories: (7) By myself → Absent and (8) By myself → By others. The last category was the “continued caregiving” group (9).

### 2.3. Statistical Analyses

A T test or analysis of variance was performed to analyze mean CES-D 10 score on the basis of caregiving-status transitions. To investigate repeat-measured participants, generalized estimating equation (GEE) analysis was applied. In addition, subgroup analysis stratified by the participants’ age, residential area, current economic activity, and number of household members was performed to evaluate their effects on the caregiver depressive symptoms. All statistical analyses were performed using SAS 9.4 (SAS Institute, Inc., Cary, NC, USA). 

### 2.4. Ethics

The KLoSA survey was approved by the state under Article 18 of the Statistics Act (Approval number 33602) and was conducted after acquiring the verbal consent of study participants. Since KLoSA data are anonymized and released to the public for scientific research, further ethical approval from institutional review board was not required to this study on the basis of Article 15.2 of the Rule of Bioethics and Safety Act in Korea.

## 3. Results

Table 1 shows the general characteristics of participants in the baseline year (2006–2008). A total of 7817 participants were included in the study, of which 3314 were men and 4503 were women. The mean CES-D 10 score was 3.17 ± 2.80 in men and 4.06 ± 2.97 in women. Among the 3314 men, 47 (1.4%) had started providing care for family members, 52 (1.6%) had stopped caregiving, 21 (0.6%) had continued caregiving, and 3194 (96.4%) were noncaregivers; their mean CES-D 10 scores were 3.87 ± 2.76, 3.96 ± 2.92, 3.38 ± 2.91, and 3.14 ± 2.79, respectively. Among the 4503 female participants, 58 (1.3%) had started caregiving, 98 (2.2%) had stopped caregiving, 42 (0.9%) had continued caregiving, and 4305 (95.6%) were noncaregivers; their mean CES-D 10 scores were 4.88 ± 3.11, 4.62 ± 3.09, 5.50 ± 2.55, and 4.03 ± 2.96, respectively.

Table 2 shows the results of the GEE model for the impact of caregiving-status transitions on depressive symptoms. Women who had become family caregivers showed more depressive symptoms (β 0.761, *p* < 0.0001) than noncaregivers did; however, this was not the case among men (β 0.330, *p* = 0.0693). After stopping family caregiving, neither men (β −0.087, *p* = 0.5851) nor women (β 0.244, *p* = 0.0626) showed differences in depressive symptoms compared to noncaregivers. In terms of continuing family caregiving, both men (β 0.616, *p* = 0.0277) and women (β 1.091, *p* < 0.0001) had higher levels of depressive symptoms than noncaregivers did.

Other covariates were associated with increased depressive symptoms, such as older age, living in an urban or rural area, being economically inactive, living without a spouse, a history of smoking (only in women), not engaging in regular physical activity, having chronic diseases, and not engaging in social activity. Participants with a higher educational level, higher income, and multiple household members (only in women), had fewer depressive symptoms than their counterparts did. 

Table 3 shows the results of the GEE model after classifying participants into nine categories on the basis of caregiving status and the presence of family members with ADL limitations. Overall, the trend of depressive symptoms based on caregiving-status transitions was similar in men and women, but women had higher beta coefficient values than men did. After yielding caregiving responsibilities to other caregivers (By myself → By others), the participants’ depressive symptoms did not differ from those of participants without family members requiring ADL assistance (β −0.044, *p* = 0.9090 in men, β 0.133, *p* = 0.7422 in women). Noncaregiving women with family members requiring ADL assistance had higher depressive-symptom scores (Absent → By others: β 0.389, *p* < 0.0001; By others → By others: β 0.494, *p* = 0.0180) than women without family members requiring ADL assistance did.

In Figure 1, associations between caregiving-status transitions and depressive symptoms, stratified by age, residential area, current economic activity, and number of household members, are presented. Among men who had started caregiving, those who were aged 75 years or older (β 0.971, *p* = 0.0045), lived in urban areas (β 0.649, *p* = 0.0371), were economically inactive (β 0.544, *p* = 0.0351), and lived alone (β 2.070, *p* = 0.0085) showed the highest beta coefficient value in each subgroup set. In the case of women who had started caregiving, those who were aged 65–74 years (β 1.144, *p* = 0.0003), lived in urban areas (β 0.823, *p* = 0.0005), were economically active (β 1.201, *p* < 0.0001), and lived alone (β 1.301, *p* = 0.0481), showed the highest beta coefficient value in each subgroup set. Participants who were most depressed by continued caregiving in each subgroup set were those aged 75 or older (β 1.606, *p* = 0.0057 in men, β 1.144, *p* = 0.0007 in women), economically inactive (β 1.151, *p* = 0.0013 in men, β 1.310, *p* < 0.0001 in women), women who lived alone (β 1.819, *p* < 0.0001), and men residing in urban areas (β 1.236, *p*=0.0373) or women residing in metropolitan areas (β 1.170, *p* = 0.0006). Among men, living alone also strengthened the association between depressive symptoms and continued caregiving, but this was not statistically significant (β 1.814, *p =* 0.2199).

## 4. Discussion

In this study, we examined the relationship between caregiving-status transitions and the magnitude of depressive symptoms among middle-aged or older Korean adults. Consistent with previous studies, our results indicated that starting (only in women) and continuing caregiving for family members with ADL limitations were associated with higher depressive-symptom scores [16,17,18,19]. Participants who had ceased caregiving the previous year did not show a difference in depressive symptoms compared to those of noncaregivers.

Theoretically, stressors for family caregivers may change over the course of caregiving: acceptance of the caregiving role at the initial stage; economic, physical, and psychological burdens from prolonged caregiving; feelings of guilt and financial concerns relating to institutionalization; and response to bereavement at the end of caregiving [11,35,36].

Previous studies showed that depressive symptoms may depend on reasons for stopping: bereavement, recovery of care recipients, or yielding caregiving responsibilities to others [18,23]. In this study, after delegating care responsibilities to other people (By myself → By others), both men and women showed no difference in depressive symptom scores compared to participants without family members requiring ADL assistance. In other words, the family caregiver’s depressive symptoms could be reduced when the care responsibility is delegated to alternative caregivers. These results justify policies that aim to expand nursing facilities and encourage the use of professional caregivers to relieve the caregiver’s burden and prevent depression. However, noncaregiving women continued to suffer from higher depressive symptoms due to the presence of family members with ADL limitations (By others → By others). Such depressive symptoms might be attributed to feelings of compassion toward relatives who are suffering, feelings of guilt and financial concerns from using a professional caregiver or nursing facilities, and anxiety about the possibility of becoming a family caregiver in the future [37,38].

In subgroup analysis, the association between family caregiving and depressive symptoms was affected by gender and other variables. Generally, the increase in depressive symptoms in family caregivers was larger among women. Depressive symptoms worsened with continued family caregiving when participants were older than 75, were economically inactive, resided in metropolitan (only in women) or urban areas, or lived alone. In other words, family caregivers with these characteristics were less resilient to the stress caused by the burden of caregiving. 

In terms of starting caregiving, men showed the same pattern as with continuing caregiving, but women’s depressive symptoms worsened when they were younger than 75, or economically active. For women without economic activity (mainly housewives), or relatively old women (older than 75) caring for their family is what they had always done, and thus, being a family caregiver may be recognized as part of their ordinary roles [39]. By contrast, younger or working women may regard being a family caregiver as a burden—an unexpected task that forces them to sacrifice their own time and career. 

On the basis of the findings of this study, appropriate support that targets caregiving status is needed to reduce family caregivers’ depressive symptoms, especially in South Korea where demands for informal care are expected to increase due to the rapidly aging population [24,25]. People who begin family caregiving can be supported through education and training programs to adjust to their new role [40]. Financial support or respite care programs can be offered to alleviate the burden of caregiving. Since family caregivers’ depressive symptoms intensify with continued caregiving, additional measures including emotional support or the provision of nursing facilities should be considered, particularly for people at high risk of developing depressive symptoms. Ultimately, comprehensive plans to support the mental health of noncaregiving family members and widows, and family caregivers, should be developed.

The current study has some limitations. Therefore, although the findings suggest an effect of caregiving transition on depressive symptoms, they should be cautiously interpreted. First, the number of participants classified as caregivers was relatively small. Second, due to limitations with the secondary data, we could not consider all possible factors that could affect a caregiver’s depressive symptoms, such as their relationship with care recipients, the severity of ADL restrictions, duration of caregiving, or coping methods. Third, we were unable to distinguish between cure and bereavement as reasons for stopping caregiving. Thus, the effects of each on depressive symptoms could have been offset. Fourth, although the CES-D 10 is a useful instrument to evaluate the magnitude of depressive symptoms, it cannot be used as a diagnostic measure of major depressive disorder by itself. Lastly, although we used longitudinal data to secure the temporal context, the data were insufficient to explain the complete causal relationship. Therefore, to understand the impact of family caregiving on mental health and establish appropriate policies for family caregivers, future studies should track full periods of care, including postcaregiving periods, and evaluate every caregiving situation that can affect a caregiver’s mental health.

## 5. Conclusions

Our findings suggest differential effects of transitions in caregiving status on depressive symptoms among older adults providing care for their family members with ADL limitations. Women who started family caregiving showed higher depressive-symptom scores than noncaregivers did. Participants who continued caregiving presented more depressive symptoms, while those who had stopped caregiving in the previous year showed no difference in depressive symptoms compared to noncaregivers. Since family caregivers are still the most important source of care for South Korean older adults, intervention strategies aimed at providing respite care should be reinforced to prevent and reduce family caregivers’ depressive symptoms.

## Figures and Tables

**Figure 1 ijerph-18-00042-f001:**
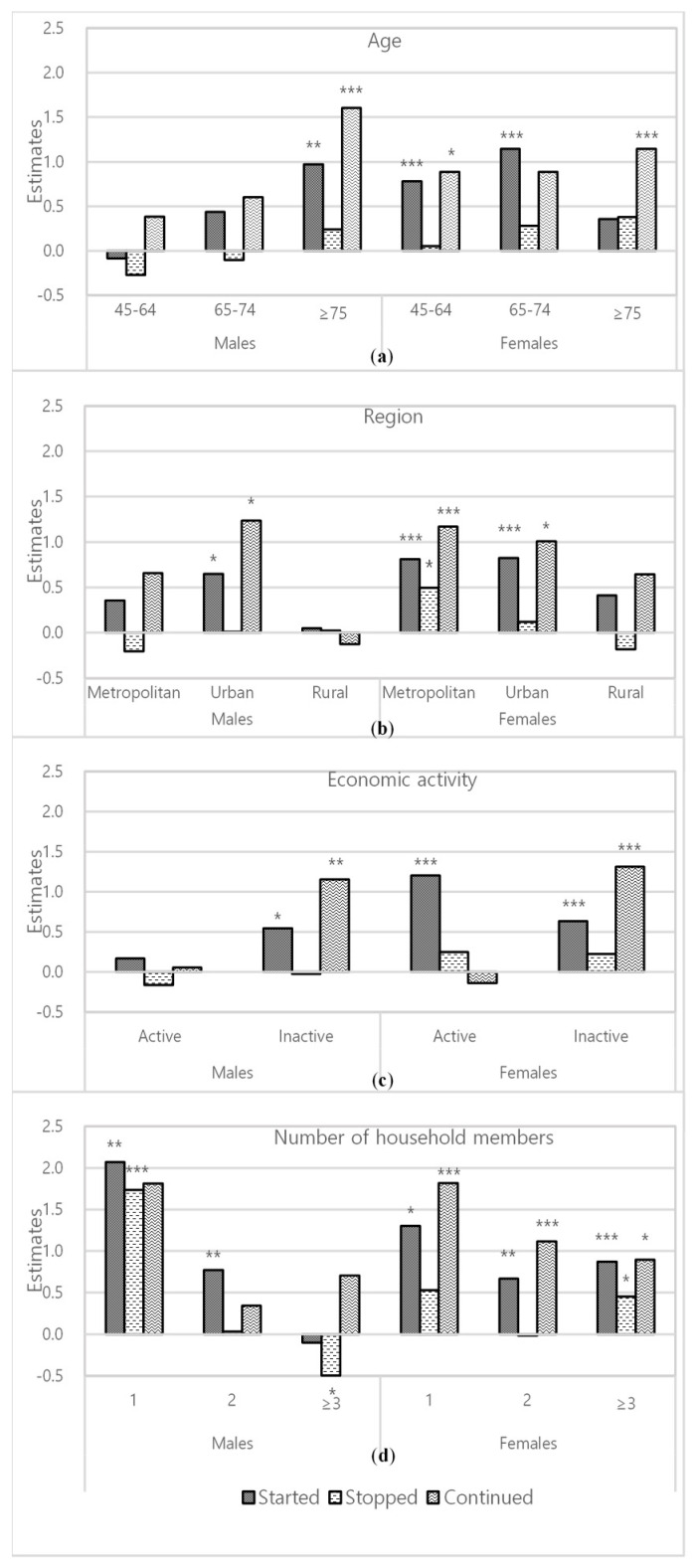
Results of subgroup analysis stratified by (**a**) age, (**b**) region, (**c**) current economic activity, and (**d**) number of household members. Each set of subgroup analysis was adjusted for other covariates (age, region, educational level, economic activity, household income, marital status, health insurance type, smoking, current alcohol drinking, regular physical activity, number of chronic diseases, and social activity). * *p* value ≤ 0.05; ** *p* value ≤ 0.01, *** *p* value ≤ 0.001.

**Table 1 ijerph-18-00042-t001:** General characteristics of study subjects (2006–2008 baseline year).

Variables (Total N = 7817)	Men				*p* Value	Women				*p* Value
	Subjects		CES-D 10 ^1^			Subjects		CES-D 10		
	N	%	Means	±S.D		N	%	Means	±S.D	
	3314	42.4	3.17	2.80		4503	57.6	4.06	2.97	
**Caregiving Status Transition**					0.0563					0.0004
Noncaregiver	3194	96.4	3.14	2.79		4305	95.6	4.03	2.96	
Started	47	1.4	3.87	2.76		58	1.3	4.88	3.11	
Stopped	52	1.6	3.96	2.92		98	2.2	4.62	3.09	
Continued	21	0.6	3.38	2.91		42	0.9	5.50	2.55	
**Age**					<0.0001					<0.0001
45–64	1876	56.6	2.66	2.56		2410	53.5	3.29	2.76	
65–74	953	28.8	3.46	2.86		1252	27.8	4.67	2.93	
≥75	485	14.6	4.58	2.99		841	18.7	5.38	2.96	
**Region**					<0.0001					<0.0001
Metropolitan	1421	42.9	2.82	2.64		2006	44.5	3.82	2.96	
Urban	1072	32.3	3.22	2.86		1394	31.0	4.08	3.02	
Rural	821	24.8	3.69	2.89		1103	24.5	4.49	2.88	
**Educational Level**					<0.0001					<0.0001
Elementary school	1023	30.9	4.02	2.96		2608	57.9	4.74	2.99	
Middle/high school	1709	51.6	2.92	2.71		1694	37.6	3.20	2.70	
College	582	17.6	2.39	2.37		201	4.5	2.65	2.52	
**Economic Activity**					<0.0001					<0.0001
Active	2010	60.7	2.62	2.52		1339	29.7	3.26	2.66	
Inactive	1304	39.3	4.02	2.98		3164	70.3	4.41	3.03	
**Equivalized Household Income**					<0.0001					<0.0001
Low	642	19.4	4.35	3.03		1225	27.2	5.21	2.98	
Middle-low	819	24.7	3.37	2.81		1065	23.7	4.22	2.92	
Middle-high	931	28.1	2.87	2.69		1114	24.7	3.55	2.85	
High	922	27.8	2.46	2.41		1099	24.4	3.17	2.69	
**Marital Status**					<0.0001					<0.0001
With spouse	3035	91.6	3.03	2.73		3030	67.3	3.63	2.85	
Without spouse	279	8.4	4.70	3.02		1473	32.7	4.96	3.00	
**Number of Household Members**					<0.0001					<0.0001
1	130	3.9	4.42	3.04		657	14.6	5.08	2.96	
2	1486	44.8	3.40	2.86		1754	39.0	4.12	2.97	
≥3	1698	51.2	2.87	2.67		2092	46.5	3.70	2.89	
**Lifetime Smoking Experience**					0.1960					<0.0001
No	1190	35.9	3.08	2.84		4304	95.6	3.99	2.95	
Yes	2124	64.1	3.21	2.77		199	4.4	5.70	3.03	
**Current Alcohol Drinking**					0.0104					0.0008
No	758	22.9	3.40	2.91		3474	77.1	4.15	2.98	
Yes	2556	77.1	3.10	2.76		1029	22.9	3.79	2.93	
**Regular Physical Activity**					<0.0001					<0.0001
Yes	1331	40.2	2.7	2.57		1494	33.2	3.37	2.78	
No	1983	59.8	3.46	2.91		3009	66.8	4.41	3.00	
**Number of Chronic Diseases ^2^**					<0.0001					<0.0001
None	1677	50.6	2.65	2.58		1959	43.5	3.29	2.75	
1	977	29.5	3.31	2.81		1403	31.2	4.22	2.98	
≥2	660	19.9	4.27	2.96		1141	25.3	5.19	2.92	
**Social Activity**					<0.0001					<0.0001
Yes	2,673	80.7	2.87	2.64		3394	75.4	3.75	2.89	
No	641	19.3	4.41	3.10		1109	24.6	5.04	3.00	

^1^ CES-D 10, 10 item Center for Epidemiological Studies Depression Scale; ^2^ chronic diseases include hypertension, diabetes, cancer, chronic obstructive pulmonary disease, liver disease, cardiac disease, psychiatric disorders, cerebrovascular disease, osteoarthritis, and rheumatoid diseases.

**Table 2 ijerph-18-00042-t002:** Association between caregiving-status transitions and depressive symptoms: results of generalized estimating equation (GEE) analysis.

	CES-D 10 Score ^1^					
Variables	Men			Women		
	β	S.E	*p* Value	β	S.E	*p* Value
**Caregiving-Status transition**						
Noncaregiver	Ref.			Ref.		
Started	0.330	0.182	0.0693	0.761	0.149	<0.0001
Stopped	−0.087	0.158	0.5841	0.244	0.13	0.0626
Continued	0.616	0.280	0.0277	1.091	0.254	<0.0001
**Age**						
45–64	Ref.			Ref.		
65–74	0.045	0.062	0.4688	0.139	0.059	0.0177
≥75	0.309	0.088	0.0005	0.298	0.079	0.0002
**Region**						
Metropolitan	Ref.			Ref.		
Urban	0.342	0.075	<0.0001	0.248	0.066	0.0002
Rural	0.160	0.080	0.0448	0.149	0.069	0.0304
**Educational Level**						
Under elementary school	Ref.			Ref.		
Middle/high school	−0.363	0.086	<0.0001	−0.368	0.072	<0.0001
College	−0.543	0.105	<0.0001	−0.374	0.134	0.0052
**Economic Activity**						
Active	Ref.			Ref.		
Inactive	0.657	0.059	<0.0001	0.467	0.051	<0.0001
**Equivalized Household Income**						
Low	Ref.			Ref.		
Middle-low	−0.054	0.067	0.4191	−0.124	0.055	0.0231
Middle-high	−0.173	0.076	0.0230	−0.304	0.060	<0.0001
High	−0.232	0.082	0.0047	−0.429	0.066	<0.0001
**Marital Status**						
With spouse	Ref.			Ref.		
Without spouse	0.912	0.108	<0.0001	0.504	0.067	<0.0001
**Number of Household Members**						
1	Ref.			Ref.		
2	0.057	0.089	0.5223	−0.137	0.060	0.0230
≥3	−0.010	0.093	0.9112	−0.225	0.063	0.0003
**Smoking**						
No	Ref.			Ref.		
Yes	−0.083	0.071	0.2430	0.620	0.140	<0.0001
**Current Alcohol Drinking**						
No	Ref.			Ref.		
Yes	0.050	0.082	0.5402	−0.060	0.062	0.3361
**Regular Physical Activity**						
Yes	Ref.			Ref.		
No	0.342	0.045	<0.0001	0.326	0.041	<0.0001
**Number of Chronic Diseases ^2^**						
None	Ref.			Ref.		
1	0.136	0.062	0.0280	0.359	0.059	<0.0001
≥2	0.462	0.077	<0.0001	0.828	0.068	<0.0001
**Social Activity**						
Yes	Ref.			Ref.		
No	0.700	0.065	<0.0001	0.435	0.050	<0.0001

^1^ CES-D 10, 10 item Center for Epidemiological Studies Depression Scale; ^2^ chronic diseases include hypertension, diabetes, cancer, chronic obstructive pulmonary disease, liver disease, cardiac disease, psychiatric disorders, cerebrovascular disease, osteoarthritis, and rheumatoid diseases.

**Table 3 ijerph-18-00042-t003:** Depressive symptoms based on caregiving status and presence of family members with ADL limitations.

Caregiving Status and Presence of Family Members with ADL ^2^ Limitations	CES-D 10 Score ^1^					
	Men			Women		
	β	S.E	*p* Value	β	S.E	*p* Value
None						
Absent ^3^ → Absent	Ref.			Ref.		
By others ^4^ → Absent	0.014	0.110	0.8968	−0.005	0.094	0.9592
Absent → By others	0.188	0.112	0.0941	0.389	0.098	<0.0001
By others → By others	0.123	0.214	0.5642	0.494	0.209	0.0180
Started						
Absent → By myself ^5^	0.328	0.193	0.0889	0.755	0.159	<0.0001
By others → By myself	0.439	0.513	0.3923	1.000	0.408	0.0142
Stopped						
By myself → Absent	−0.083	0.173	0.6332	0.282	0.137	0.0395
By myself → By others	−0.044	0.382	0.9090	0.133	0.403	0.7422
Continued	0.625	0.280	0.0256	1.115	0.254	<0.0001

^1^ CES-D 10, 10 item Center for Epidemiological Studies Depression Scale. ^2^ ADL, activities of daily living. ^3^ “Absent” group includes participants without family members having ADL limitations. ^4^ “By others” group includes participants having family members with ADL limitations who were cared for by other caregivers. ^5^ “By myself” group includes family caregivers.

## Data Availability

This study used publicly available datasets from the Korea Longitudinal Study of Aging, 2006, 2008, 2010, 2012, 2014, 2016, and 2018, Korea Employment Information Service. This data can be found here: https://survey.keis.or.kr/eng/klosa/klosa01.jsp.

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
