# Peer review of "The Impact of Transitions in Caregiving Status on Depressive Symptoms among Older Family Caregivers: Findings from the Korean Longitudinal Study of Aging"

_ijerph, 2020, doi:10.3390/ijerph18010042_

Round 1

Reviewer 1 Report

Depressive symptomatology is a variable that has strong implications for the health of carers. The work shown below benefits from a large sample, whose data were collected over an equally large period. The location of the work (Korea) also helps to enable further comparative studies between different countries. The data shown in this study justify the need for intervention programmes and strategies. 

The following are some aspects and recommendations in case they may be useful:

INTRODUCTION

Considering the sample size and all the aspects that can be studied in this research, it would be interesting to break down the main objective, set out in the last paragraph of the introduction, into specific objectives, while of course retaining this general objective in the wording.

With regard to the format, it is recommended that the style of quotation be revised, leaving a space between the word and the beginning of the brackets.

The information provided in the introduction is interesting and helps to locate the object of study.

MATERIALS AND METHOD

How the interventions for information collection were structured is expressed in great detail in the sub-section "Study sample". However, more information should be added regarding socio-demographic variables (mean age and standard deviation, frequencies and percentages of the gender variable, etc.).

Lines 87-89 show that the "CES-D 10" instrument is valid and reliable in other studies. It is recommended that these results be described in more detail, specifying, for example, the Cronbach's Alpha value if this is the index used. Also the value of internal consistency in this research could appear. With regard to the CES-D 10, it is also recommended to include some example items.

It would be advisable to unify section 2.2 and 2.3 in a single section called "Instruments" if the authors consider it also accurate. Similarly, if the section on "covariates" refers to the socio-demographic variables, etc. described to the sample, it would perhaps be more recommended that this information appears when describing the participants. 

RESULTS

The results are widely described. The internal structure followed by the results could be used to further define the specific objectives discussed in the introductory section of this review.

DISCUSSION

Adequate information in accordance with the proposed results. It is recommended to make a greater comparison with the references seen in the introduction. In the introduction 22 bibliographical references have been analyzed and none of them appear in the theoretical framework. In order to better link the results of the study to the theoretical framework, more citations should be introduced from the introduction.

The limitations are very well developed and described. It would be interesting to place greater emphasis on future lines of research.

CONCLUSIONS

Appropriate. They describe the essential aspects of the study and make it clear what the results are. 

In short, it is a very interesting study that helps to highlight a relevant social problem. The structure of the studio is appropriate. The research shows adequate data analysis and results.

Author Response

Response to #Reviewer 1’s comments

Dec 09, 2020

We would like to give sincere appreciation for your review. The manuscript (Manuscript-ID: ijerph-1021081) entitled, “The impact of transitions in caregiving status on depressive symptoms among older family caregivers: findings from the Korean Longitudinal Study of Aging”, has been revised, according to your advice. A detailed point-by-point response to the comments follows below,

INTRODUCTION

Point 1: Considering the sample size and all the aspects that can be studied in this research, it would be interesting to break down the main objective, set out in the last paragraph of the introduction, into specific objectives, while of course retaining this general objective in the wording.

Response 1: Thank you for the comment. We added specific objectives of this study in the last paragraph of the introduction.

Page 2, line 66-73; This study, therefore, investigates the magnitude of depressive symptoms among caregivers, based on caregiving status transition, using large-sized national longitudinal survey data. In Addition, subgroup analyses were conducted to answer the following questions. After relinquishing caregiving responsibilities to others, how would caregiver’s depressive symptoms change? What is the effect of gender and other socio-demographic factors such as age, residential area, current economic activity, and number of household members, on the association between caregiving status transitions and depressive symptoms?

Point 2: With regard to the format, it is recommended that the style of quotation be revised, leaving a space between the word and the beginning of the brackets.

The information provided in the introduction is interesting and helps to locate the object of study.

Response 2: Sorry for the miswriting. We thoroughly reviewed again and revised all citations, as per reviewer’s comment.

MATERIALS AND METHOD

Point 3: How the interventions for information collection were structured is expressed in great detail in the sub-section "Study sample". However, more information should be added regarding socio-demographic variables (mean age and standard deviation, frequencies and percentages of the gender variable, etc.).

Response 3: Thank you for your comments. We added more information about participant’s gender and age in the subsection “Study sample” of the Materials and Methods.

Page 2, line 88-99; All data were stratified by sex. Of the total 7,817 participants, 3,314 (42.4%) were men and 4,503 (57.6%) were women. We included age (45–64, 65–74, 75 or more), area of residence (metropolitan, urban, rural), education level (elementary school or lower, middle or high school, college or higher), economic activity (active, inactive), equivalized household income (divided into quartiles), marital status (with spouse, without spouse), number of household members (1, 2, 3 or more), and participation in social activity (yes, no) as demographic and socioeconomic variables. The mean and standard deviation of the participant’s ages were 65.1±10.4 for men and 65.8±11.0 for women. Lifetime smoking experience (yes, no), current alcohol drinking (yes, no), number of chronic diseases (none, 1, 2 or more), and participation in regular physical activity more than once a week (yes, no) were included as health-related factors. Chronic diseases include hypertension, diabetes, cancer, chronic obstructive pulmonary disease, liver disease, cardiac disease, psychiatric disorders, cerebrovascular disease, osteoarthritis, and rheumatoid diseases.

Table 1; Variables → Variables (Total N=7,817)

Table 1; % (percentage) of men; 100 → 42.4

Table 1; % (percentage) of women; 100 → 57.6

Point 4: Lines 87-89 show that the "CES-D 10" instrument is valid and reliable in other studies. It is recommended that these results be described in more detail, specifying, for example, the Cronbach's Alpha value if this is the index used. Also the value of internal consistency in this research could appear. With regard to the CES-D 10, it is also recommended to include some example items.

Response 4: In this study, the internal consistency of CES-D 10 was acceptable (Cronbach α=0.81). And we also added several example items of CES-D 10.

Page 3, line 102-108; The CES-D 10 scores "0” or "1” for each of the 10 items, and the total score ranges from "0” to "10”. Example items are here: “During last week, I felt depressed”, “During last week, I felt that everything I did was an effort” and respondent can answer dichotomously. Higher overall scores suggest more severe depressive symptoms. The scale’s validity and reliability as a screening instrument for depressive symptoms in older adults have been verified in previous studies [32, 33]. The internal consistency of the CES-D 10 was satisfactory (Cronbach α=0.81) in this study.

Point 5: It would be advisable to unify section 2.2 and 2.3 in a single section called "Instruments" if the authors consider it also accurate. Similarly, if the section on "covariates" refers to the socio-demographic variables, etc. described to the sample, it would perhaps be more recommended that this information appears when describing the participants.

Response 5: Thank you for the comments. We moved the information about covariates in the latter part of subsection “Study sample”. We unified subsection 2.2 “Depressive symptoms” and subsection 2.3 “Transitions in caregiving status” into a single subsection 2.2 “Instruments”.

Page 2-3, line 76-99;

2.1 Study sample

The study data were collected from the Korea Longitudinal Study of Aging (KLoSA) for the years 2006, 2008, 2010, 2012, 2104, 2016, and 2018. The KLoSA is a nationally representative survey, conducted every two years by the Korea Employment Information Service (KEIS). It aims to generate basic data needed to devise and implement policies that address emerging trends in the process of population aging. Details on the KLoSA can be found on the KEIS webpage [29]. In 2006, the original panel sample was composed of 10,254 adults aged 45 years and over (born in 1961 or earlier), and who resided in South Korea. The retention rates of the survey sample were 86.6%, 81.7%, 80.1%, 80.4%, 79.6%, and 78.8%, for the 2008, 2010, 2012, 2014, 2016, and 2018 surveys, respectively [30]. Participants who were already taking antidepressants at the time of survey, or those who had any incomplete data, were excluded. The total number of participants was 7,817 in the final sample of the baseline study year from 2006 to 2008.

All data were stratified by sex. Of the total 7,817 participants, 3,314 (42.4%) were men and 4,503 (57.6%) were women. We included age (45–64, 65–74, 75 or more), area of residence (metropolitan, urban, rural), education level (elementary school or lower, middle or high school, college or higher), economic activity (active, inactive), equivalized household income (divided into quartiles), marital status (with spouse, without spouse), number of household members (1, 2, 3 or more), and participation in social activity (yes, no) as demographic and socioeconomic variables. The mean and standard deviation of the participant’s ages were 65.1±10.4 for men and 65.8±11.0 for women. Lifetime smoking experience (yes, no), current alcohol drinking (yes, no), number of chronic diseases (none, 1, 2 or more), and participation in regular physical activity more than once a week (yes, no) were included as health-related factors. Chronic diseases include hypertension, diabetes, cancer, chronic obstructive pulmonary disease, liver disease, cardiac disease, psychiatric disorders, cerebrovascular disease, osteoarthritis, and rheumatoid diseases.

Page 3, line 100; 2.2 Instruments

RESULTS

The results are widely described. The internal structure followed by the results could be used to further define the specific objectives discussed in the introductory section of this review.

DISCUSSION

Point 6: Adequate information in accordance with the proposed results. It is recommended to make a greater comparison with the references seen in the introduction. In the introduction 22 bibliographical references have been analyzed and none of them appear in the theoretical framework. In order to better link the results of the study to the theoretical framework, more citations should be introduced from the introduction.

Response 6: We introduced more citations from the introduction into the discussion, to explain our findings. To this end, several references have been added.

Page 9, line 191-194; Consistent with previous studies, our results indicated that starting (only in women) and continuing caregiving for family members with ADL limitations were associated with higher depressive symptom scores [16-19].

Page 9, line 200-201; Previous studies have shown that depressive symptoms may depend on reasons for stopping: bereavement, recovery of care-recipients, or yielding caregiving responsibilities to others [18, 23].

Page 10, line 226-228; Based on the findings of this study, appropriate support targeting caregiving status is needed to reduce family caregivers’ depressive symptoms, especially in South Korea where demands for informal care is expected to increase due to rapidly aging population [24, 25].

Page 12, line 315-319;

18.          Kaufman, J. E.; Lee, Y.; Vaughon, W.; Unuigbe, A.; Gallo, W. T. Depression associated with transitions into and out of spousal caregiving. Int. J. Aging Hum. Dev. 2019, 88, 127-149.

19.          Roth, D. L.; Haley, W. E.; Rhodes, J. D.; Sheehan, O. C.; Huang, J.; Blinka, M. D.; Yuan, Y.; Irvin, M. R.; Jenny, N.; Durda, P. Transitions to family caregiving: Enrolling incident caregivers and matched non-caregiving controls from a population-based study. Aging Clin. Exp. Res. 2019, 1-10.

Page 12, line 327-329;

23.          Perone, A. K.; Dunkle, R. E.; Feld, S.; Shen, H.-W.; Kim, M. H.; Pace, G. T. Depressive Symptoms among Former Spousal Caregivers: Comparing Stressors, Resources, and Circumstances of Caregiving Cessation among Older Husbands and Wives. J. Gerontol. Soc. Work 2019, 62, 682-700.

Point 7: The limitations are very well developed and described. It would be interesting to place greater emphasis on future lines of research.

Response 7: As follows, we changed the sentence regarding future research with more emphasis.

Page 10, line 246-249; Therefore, to understand impact of family caregiving on mental health, and to establish appropriate policies for family caregivers, future studies should track full periods of care including post-caregiving periods, and have to evaluate every caregiving situations that can affect caregiver’s mental health.

CONCLUSIONS

Appropriate. They describe the essential aspects of the study and make it clear what the results are.

In short, it is a very interesting study that helps to highlight a relevant social problem. The structure of the studio is appropriate. The research shows adequate data analysis and results.

Reviewer 2 Report

Dear Authors,

I read with interest and attention the manuscript you submitted for publication. 

Here are my comments and suggestions:

Major comments:

First of all, I found a number of unclear terminological points and some nosographic inaccuracies : depression or depressive symptoms ? Major depressive disorder or reactive depression? These points should be better clarified in the Introduction section. Please, what do you want to write and discuss about ? If You want to write about depressive symptoms (as I seemed reading Your manuscript), the title must be properly modified. 

You evaluated participants' depressive symptoms using the CES-D10. As You know, this scale (like others....) can be useful for classification purposes but do not offer a key to an unambiguous diagnosis, only attainable through clinical assessment. Indeed, clinical assessment is the agreed diagnostic gold standard.  In short, the CES-D10 is only a screening instrument, as You correctly wrote in lines 87 and 88.  This could be a limit, and should be properly discussed.   

In Materials and Methods, You wrote that participants who were already taking antidepressant at the time of survey, were excluded. Could this exclusion criterion justify the low mean CES-D 10 total score ? Please, discuss this point.  

Minor point:

Please, consider to include in the bibliography more articles published in the last years. 

Author Response

Response to #Reviewer 2’s comments

Dec 09, 2020

We would like to give sincere appreciation for your review. The manuscript (Manuscript-ID: ijerph-1021081) entitled, “The impact of transitions in caregiving status on depressive symptoms among older family caregivers: findings from the Korean Longitudinal Study of Aging”, has been revised, according to your advice. A detailed point-by-point response to the comments follows below,

Point 1: First of all, I found a number of unclear terminological points and some nosographic inaccuracies : depression or depressive symptoms ? Major depressive disorder or reactive depression? These points should be better clarified in the Introduction section. Please, what do you want to write and discuss about ? If You want to write about depressive symptoms (as I seemed reading Your manuscript), the title must be properly modified.

Response 1: Sorry for the miswriting. We thoroughly reviewed manuscript and clarified all terms regarding major depressive disorder or depressive symptoms.

Page 1, line 3(Title); depressive symptoms among older family caregivers

Page 1, line 31-39; Major depressive disorder, also known as depression, is a serious, and increasingly common, mental problems worldwide, with more than 160 million people affected in 2017 [1]. In South Korea, the number of individuals with major depressive disorder was 908,000 (1.78%) in 2019, an increase from 788,000 (1.69%) in 2009 [2]. Moreover, much more South Korean, from 25.3% to 38.9% of total population, were reported to having depressive symptoms [3].

An individual’s probability of suffering from an episode of major depression is affected by many factors, including predisposing genetic influences, exposure to traumatic events, adverse social factors, a prior history of major depressive disorder, and recent stressful life events and difficulties [4].

Page 1-2, line 41-43; Especially, family members who act as caregivers for patients with limitations in activities of daily living (ADL) are at a higher risk of having depressive symptoms [5-7].

Page 9, line 174-176; In Figure 1, associations between caregiving status transitions and depressive symptoms, stratified by age, residential area, current economic activity, and number of household members, are presented.

Page 9, line 201-204; In this study, after delegating care responsibilities to other people (By myself → By others), both men and women showed no difference in depressive symptom scores compared to participants without family members requiring ADL assistance.

Page 9, line 207-209; However, non-caregiving women continued to suffer from higher depressive symptoms due to the presence of family members with ADL limitations (By others → By others).

Page 9, line 231-233; Since family caregivers’ depressive symptoms intensify with continued caregiving, additional measures—including emotional support or provision of nursing facilities—should be considered, particularly for people at high risk of developing depressive symptoms.

Page 9, line 237-241; First, the number of participants classified as caregivers was relatively small. Second, due to limitations with the secondary data, we could not consider all possible factors that can affect a caregiver’s depressive symptoms, such as their relationship with the care recipients, severity of ADL restriction, duration of caregiving, or coping methods.

Point 2: You evaluated participants' depressive symptoms using the CES-D10. As You know, this scale (like others....) can be useful for classification purposes but do not offer a key to an unambiguous diagnosis, only attainable through clinical assessment. Indeed, clinical assessment is the agreed diagnostic gold standard.  In short, the CES-D10 is only a screening instrument, as You correctly wrote in lines 87 and 88.  This could be a limit, and should be properly discussed.

Response 2: Thank you for the comments. We added the limitations of CES-D 10 in the last paragraphs of Discussion as follows.

Page 11, line 243-244; Fourth, although the CES-D 10 is a useful instrument to evaluate the magnitude of depressive symptoms, it cannot be used as a diagnostic measure of major depressive disorder by itself.

Point 3: In Materials and Methods, You wrote that participants who were already taking antidepressant at the time of survey, were excluded. Could this exclusion criterion justify the low mean CES-D 10 total score ? Please, discuss this point.

Response 3: Thank you for the comments. However, in this study, we used the Boston form CES-D 10, which scored its items dichotomously. Thus, the maximum score of CES-D 10 was 10, and mean of total CES-D 10 score in our study sample was not thought to be low (men: 3.17±2.80, women:4.06±2.97 in the 2006–2008 baseline study year). Other studies with the same data also showed similar scores when using the binary responses version CES-D 10.

e.g.) Kim, S.; Subramanian, S. Income volatility and depressive symptoms among elderly Koreans. Int. J. Environ. Res. Public Health 2019, 16, 3580. (mean CES-D 10 total score: 4.00±3.00 among subjects aged 60 or older, in 2006 survey year)

Point 4: Please, consider to include in the bibliography more articles published in the last years.

Response 4: We cited several articles published in the last years as follows.

Page 12, line 315-319;

18.          Kaufman, J. E.; Lee, Y.; Vaughon, W.; Unuigbe, A.; Gallo, W. T. Depression associated with transitions into and out of spousal caregiving. Int. J. Aging Hum. Dev. 2019, 88, 127-149.

19.          Roth, D. L.; Haley, W. E.; Rhodes, J. D.; Sheehan, O. C.; Huang, J.; Blinka, M. D.; Yuan, Y.; Irvin, M. R.; Jenny, N.; Durda, P. Transitions to family caregiving: Enrolling incident caregivers and matched non-caregiving controls from a population-based study. Aging Clin. Exp. Res. 2019, 1-10.

Page 12, line 327-329;

23.          Perone, A. K.; Dunkle, R. E.; Feld, S.; Shen, H.-W.; Kim, M. H.; Pace, G. T. Depressive Symptoms among Former Spousal Caregivers: Comparing Stressors, Resources, and Circumstances of Caregiving Cessation among Older Husbands and Wives. J. Gerontol. Soc. Work 2019, 62, 682-700.

Page 13, line 366-368;

40.          Lee, K.; Puga, F.; Pickering, C. E.; Masoud, S. S.; White, C. L. Transitioning into the caregiver role following a diagnosis of Alzheimer’s disease or related dementia: A scoping review. Int. J. Nurs. Stud. 2019, 96, 119-131.

Reviewer 3 Report

This is an excellent manuscript describing longitudinal effects of caregiving status on depressive symptoms among older people caring for family with limitations in activities of daily living. The writing is clear and concise, and the methods are appropriate. I only have minor comments to improve the manucsript:

Line 56 and 58: Please consider refraining from use of the word elderly. As people are living longer, and working longer, it can be considered to be an offensive word to some people. “Older adults” is fine. Relevant for rest of paper.

Line 114: Is drinking referring to currently drinking or previous drinking? I also suggest this be renamed to Alcohol. Relevant for Table 1 and rest of paper.

Line 115: Please define regular physical activity.

Author Response

Response to #Reviewer 3’s comments

Dec 09, 2020

We would like to give sincere appreciation for your review. The manuscript (Manuscript-ID: ijerph-1021081) entitled, “The impact of transitions in caregiving status on depressive symptoms among older family caregivers: findings from the Korean Longitudinal Study of Aging”, has been revised, according to your advice. A detailed point-by-point response to the comments follows below,

Point 1: Line 56 and 58: Please consider refraining from use of the word elderly. As people are living longer, and working longer, it can be considered to be an offensive word to some people. “Older adults” is fine. Relevant for rest of paper.

Response 1: Thank you for the comments. The terms “elderly” have been changed to “older adults” as follows.

Page 2, line 59-61; In South Korea, particularly, the number of older adults with ADL limitations is expected to increase due to a rapid aging population [24]. Moreover, in South Korea, the majority of older adults are cared for by their family members, who are also aged themselves [25].

Point 2: Line 114: Is drinking referring to currently drinking or previous drinking? I also suggest this be renamed to Alcohol. Relevant for Table 1 and rest of paper.

Point 3: Line 115: Please define regular physical activity.

Response 2 and 3: We clarified the term “drinking” as “current alcohol drinking”. Also, definition of the term “regular physical activity” was clarified as the “participation in regular physical activity more than once a week” as follows.

Page 3, line 94-97; Lifetime smoking experience (yes, no), current alcohol drinking (yes, no), number of chronic diseases (none, 1, 2 or more), and participation in regular physical activity more than once a week (yes, no) were included as health-related factors.

Table 1; Drinking Current alcohol drinking

Table 2; Drinking Current alcohol drinking

Figure 1, footnotes; Each set of subgroup analysis was adjusted for other covariates (age, region, educational level, economic activity, household income, marital status, number of household members,  smoking, current alcohol drinking, regular physical activity, number of chronic diseases, and social activity)

Round 2

Reviewer 2 Report

Dear Author,

I read with attention the newer, revised version of our manuscript. 

Some of my suggestions were satisfatorily met. 

Unfortunately, Your level of knowledge of what depression is, seemed quite modest. Indeed, a persistent therminological confusion was present (please, see line 31, 36 and so on....).

Maybe a psychiatrist or a neurologist should be involved among the authors. 

Obviously, the Academic Editor or the Editor in chief will consider whether this my suggestion can improve the quality and the scientific soundness of Your manuscript.